# Recurrence rates after surgical removal of oral leukoplakia—A prospective longitudinal multi-centre study

Jonas Sundberg[1]ᵒ, Magdalena Korytowska[2]ᵒ, Erik Holmberg[3], John Bratel[4], Mats Wallström[5], Ebba Kjellström[6], Johan Blomgren[7], Anikó Kovács[8], Jenny Öhman[1], Lars Sand[9], Jan-Michaél Hirsch[6], Daniel Giglio[3], Göran Kjeller[5,10], Bengt Hasséus[1,4]*

1 Department of Oral Medicine and Pathology, Institute of Odontology, The Sahlgrenska Academy, University of Gothenburg, Gothenburg, Sweden, 2 Clinic of Orofacial Medicine, NÄL Hospital, Region Västra Götaland, Trollhättan, Sweden, 3 Department of Oncology, Institute of Clinical Sciences, The Sahlgrenska Academy, University of Gothenburg, Gothenburg, Sweden, 4 Clinic of Oral Medicine, Region Västra Götaland, Gothenburg, Sweden, 5 Clinic of Oral and Maxillofacial Surgery, Region Västra Götaland, Gothenburg, Sweden, 6 Department of Surgical Sciences, Oral and Maxillofacial Surgery, Uppsala University, Uppsala, Sweden, 7 Clinic of Oral Medicine, Sahlgrenska University Hospital/East, Region Västra Götaland, Gothenburg, Sweden, 8 Department of Clinical Pathology, Sahlgrenska University Hospital, Gothenburg, Sweden, 9 Department of Oral Biology, Faculty of Odontology, University of Oslo, Oslo, Norway, 10 Department of Oral and Maxillofacial Surgery, Institute of Odontology, The Sahlgrenska Academy, University of Gothenburg, Gothenburg, Sweden

ᵒ These authors contributed equally to this work.
* bengt.hasseus@odontologi.gu.se

**Data Availability Statement:** There are ethical and legal restrictions on sharing our data set according to Swedish law. Study data contain sensitive patient information that can be connected to

## Abstract

Oral leukoplakia (OL) is a potentially malignant oral disorder. The Gold Standard treatment is to remove surgically the OL. Despite optimal surgery, the recurrence rates are estimated to be 30%. The reason for this is unknown. The aim of this study was to investigate the clinical factors that correlate with recurrence after surgical removal of OL. In a prospective study data were collected from 226 patients with OL. Forty-six patients were excluded due to incomplete records or concomitant presence of other oral mucosal diseases. Overall, 180 patients proceeded to analysis (94 women and 86 men; mean age, 62 years; age range, 28–92 years). Clinical data, such as gender, diagnosis (homogeneous/non-homogeneous leukoplakia), location, size, tobacco and alcohol use, verified histopathological diagnosis, and clinical photograph, were obtained. In patients who were eligible for surgery, the OL was surgically removed with a margin. To establish recurrence, a healthy mucosa between the surgery and recurrence had to be confirmed in the records or clinical photographs. Statistical analysis was performed with the level of significance set at P<0.05. Of the 180 patients diagnosed with OL, 57% (N = 103) underwent surgical removal *in toto*. Recurrence was observed in 43 OL. The cumulative incidence of recurrence of OL was 45% after 4 years and 49% after 5 years. Fifty-six percent (N = 23) of the non-homogeneous type recurred. Among snuff-users 73% (N = 8) cases of OL recurred. A non-homogeneous type of OL and the use of snuff were significantly associated with recurrence after surgical excision (P = 0.021 and P = 0.003, respectively). Recurrence was also significantly associated with cancer transformation (P<0.001). No significant differences were found between recurrence and any of the following: dysplasia, site of lesion, size, multiple vs. solitary OL, gender,

individual patients. It may be possible to identify and connect personal information from the data set, even though data is de-identified. All relevant data within the paper are at group level. Data cannot be shared publicly because all data are research data and belongs to University of Gothenburg. Data are available upon request based on ethical and legal restrictions from GU server, University of Gothenburg. Institutional Data Access Contact: Associate Professor Bengt Hasséus, Department of Oral medicine and Pathology, Institute of Odontology, the Sahlgrenska Academy, University of Gothenburg, Sweden, at bengt. hasseus@gu.se or the Head of the Institute of Odontology, Professor Peter Lingström at peter. lingstrom@odontologi.gu.se for researchers who meet the criteria for access to confidential data.

**Funding:** This study was supported by grants from: The Healthcare Board, Region Västra Götaland (Hälso- och sjukvårdsstyrelsen), TUA Research Funding; The Sahlgrenska Academy at the University of Gothenburg/Region Västra Götaland, Assar Gabrielsson Foundation, Adlerbertska Foundation, Swedish Dental Society, and Gothenburg Dental Society, Sweden. Grants: The Healthcare Board, Region Västra Götaland (Hälso- och sjukvårdsstyrelsen), TUA Research Funding, https://www.researchweb.org/is/tuagbg, Grants number: TUAGBG-620871, VGFOUREG-647771, Initials of the authors who received the grant: BH; The Sahlgrenska Academy at the University of Gothenburg/Region Västra Götaland, Assar Gabrielsson Foundation, http://www.agfond. se, Initials of the authors who received the grant: BH; The Sahlgrenska Academy at the University of Gothenburg/Region Västra Götaland, Adlerbertska Foundation, https://www.gu.se/forskning/ stipendier/gustipendier/adlerbertska_ forskningsstiftelsen, Initials of the authors who received the grant: JS; Swedish Dental Society, Sweden, https://tandlakarforbundet.se/forskning/ vetenskapliga-fonder/, Grants number: 1444, Initials of the authors who received the grant: JS; Gothenburg Dental Society, Sweden, https:// goteborgstandlakaresallskap.nu, Grants number: -, Initials of the authors who received the grant: JS. The funders had no role in study design, data collection and analysis, decision to publish, or preparation of the manuscript.

**Competing interests:** The authors have declared that no competing interests exist.

age, use of alcohol or smoking. In conclusion, clinical factors that predict recurrence of OL are non-homogeneous type and use of snuff.

## Introduction

Oral leukoplakia (OL) is a potentially malignant oral disorder (PMOD) that sometimes transforms into oral squamous cell carcinoma (OSCC). OL, which is defined as "a white plaque of questionable risk, (other) known diseases or disorders that carry no increased risk of cancers having been excluded" [1], is one of the most frequent PMODs seen in the oral cavity. The global prevalence of OL is approximately 2.6% [2].

Clinically, OL may present as homogeneous or non-homogeneous (Fig 1A and 1B). Homogeneous leukoplakia, which is the most common form, is manifested as a flat and uniform white plaque with a smooth surface and well-defined margins. Non-homogeneous OL appears as a white plaque and areas of erythema accompanied by areas that contain nodules and/or verrucous parts with ill-defined margins [3].

The pathogenesis of OL is unknown. However, OL has been associated with several pre-disposing factors, such as tobacco use and excessive consumption of alcohol [4]. Some studies have pointed to a possible correlation between OL and human papillomavirus (HPV) infection of the oral epithelium [5]. Different genetic alterations, including changes in genes that regulate the genomic stability, cell cycle, cytoskeleton, angiogenesis, and apoptosis, in the OL epithelium have been suggested to be drivers of tumorigenesis [6–8].

Warnakulasuriya et al. estimated the overall malignant transformation rate for OL as 3.5%, with a wide range of 0.13%–34.0% [9]. On an annual basis, the transformation rate for OL is estimated to be in the range of 1.0%–2.6% [10, 11]. Risk factors that have been reported to be associated with increased malignant transformation are: clinical type (non-homogeneous), female gender, lesion size, and presence of epithelial dysplasia [4].

Treatment modalities include continuous monitoring to tobacco/alcohol cessation, pharmacological treatment, and surgery with scalpel or laser or cryosurgery [12, 13]. Surgical excision, if possible, and surveillance are considered to be the Gold Standard for managing OL. However, data retrieved from observational studies do not support the notion that surgery reduces the risk of either recurrence or malignant transformation [11, 14, 15]. If excision is not feasible, the currently available management options are multiple surgical incision biopsies and surveillance or surveillance alone [12, 16–18].

Therefore, despite surgical intervention, cancer transformation occurs in 3–11% of cases at the site of the excised lesion [11, 15, 16, 18]. Even after radical surgical removal, OL recurrence rates are reported to be in the range of 13–42% [11, 15]. The reason for this is unknown, and the recurrence patterns of OL have been found to be independent of the intervention procedure [11, 19]. However, the results of those studies are difficult to compare, due to differences in the study designs, inclusion and exclusion criteria, treatment interventions, surgical techniques, and follow-up times.

The majority of studies that have monitored recurrence rates after surgical removal of OL have had a retrospective design. Therefore, the aim of the present prospective study was to characterise the clinical, laboratory, and anamnestic factors that correlated with the recurrence of OL after surgical removal.

## Patients and methods

In a prospective, on-going, longitudinal, multi-centre study (ORA-LEU-CAN Study) conducted in Sweden, 226 patients have been examined at five sites in: Gothenburg (Clinic of Oral

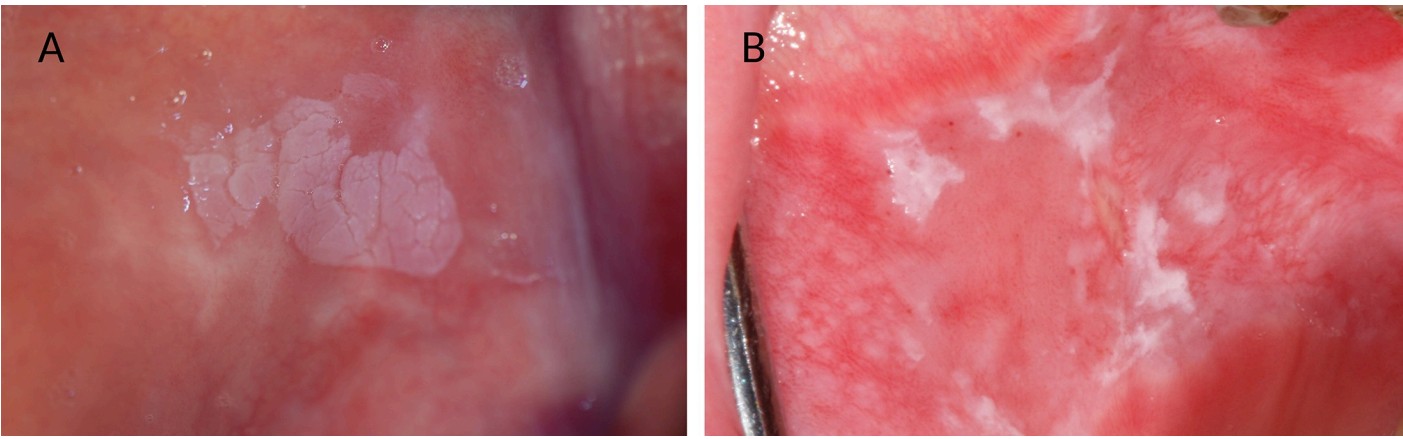

**Fig 1. Clinical presentation of homogeneous leukoplakia.** (A) and non-homogeneous leukoplakia (B) in the buccal mucosa.

Medicine, Clinic of Oral and Maxillofacial Surgery and Clinic of Oral Medicine, Sahlgrenska University Hospital/East), Trollhättan (Clinic of Orofacial Medicine, NÄL Hospital), and Uppsala (Department of Surgical Sciences, Oral and Maxillofacial Surgery, Uppsala University Hospital).

In the present study, patients were included during the period January 2011 to December 2018. Patients were referred to the participating centres from general dental practitioners, general medical practitioners and Ear-Nose-Throat specialists and subjected to analysis. A patient needed to be followed for at least 6 months in the ORA-LEU-CAN Study to be included in the present study. The inclusion criterion was a clinically verified diagnosis of OL. The participating centres defined OL in accordance with the WHO criterion; as a "white plaque of questionable risk, (other) known diseases or disorders that carry no increased risk of cancers having been excluded[1]".

Anamnestic data, including medical history, gender, age, medication, tobacco habits and alcohol habits, were collected. The items of clinical information registered were clinical diagnosis (homogeneous or non-homogeneous OL), localisation, lesion size, and multiple or single lesion. Clinical photographs and the results of histopathological examinations were collected. Epithelial dysplasia was histopathologically scored according to the World Health Organization (WHO) classification scale [20]. For the analysis a binary dysplasia scale was used, i.e., no dysplasia or dysplasia.

The follow-up intervals during the 5-year study period were every third month in the first 2 years and every sixth month in the subsequent 3 years.

All the patients received treatment according to the standard of care at each participating centre, which comprised surgical removal of OL (when possible) and counselling on tobacco and alcohol habits, when appropriate [3].

The clinical diagnoses for all the patients were re-reviewed by two specialists in oral medicine. When there was a difference of opinion regarding the diagnosis, a discussion was initiated until consensus was reached. Following the re-review process, of the 226 recruited patients, 46 were excluded due to: revision of the diagnosis made at inclusion (N = 11 had OSCC at inclusion and N = 20 had concomitant other oral mucosal disorders); incomplete records (N = 13); or OSCC developed at another site in the oral cavity (N = 2). Thus, 180 patients with the clinical diagnosis of OL proceeded to analysis (Fig 2). The patients' characteristics are listed in Table 1.

The OL was removed with a 2-mm clinical margin using conventional scalpel surgery and sent for histopathological analysis.

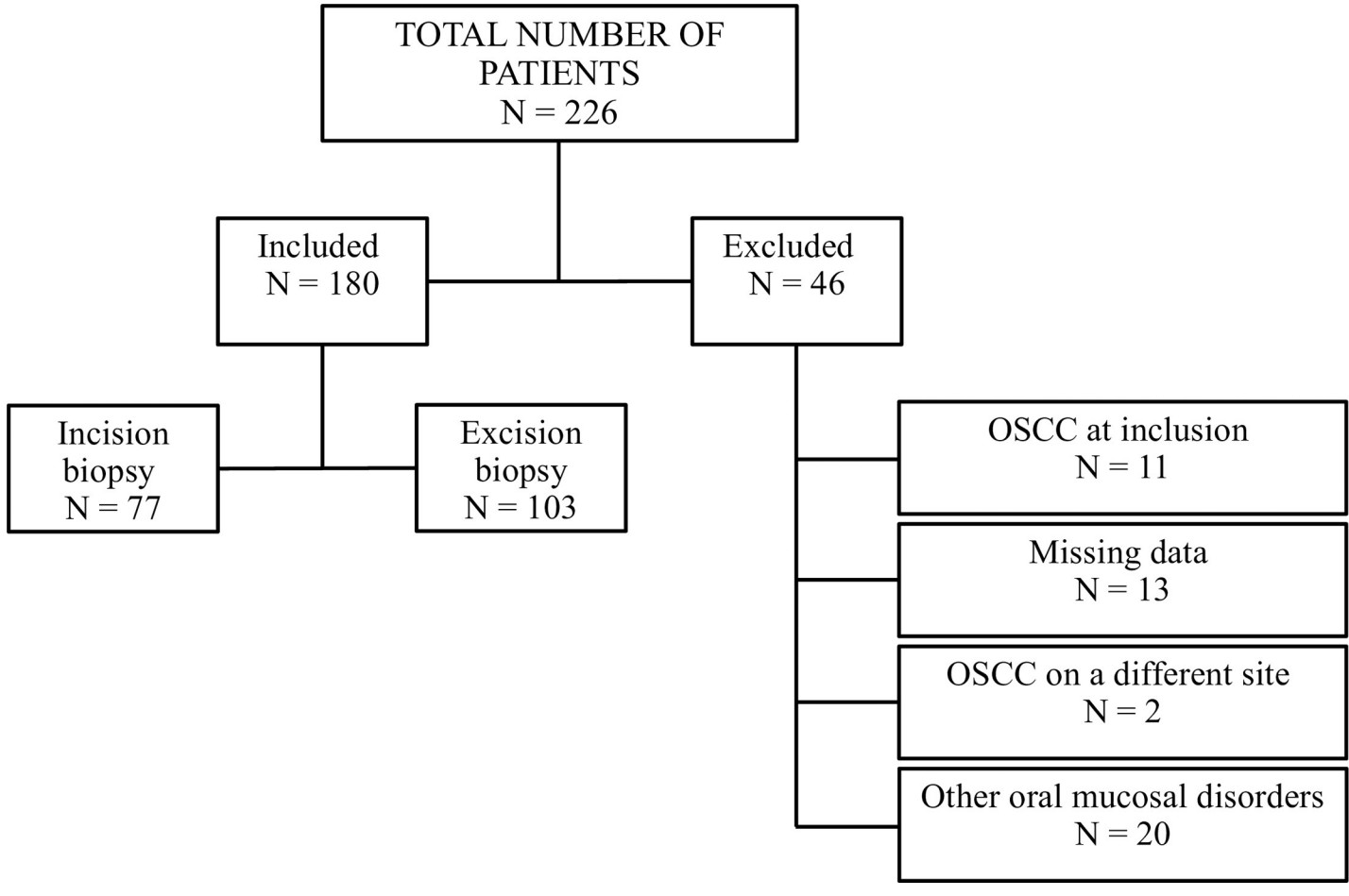

**Fig 2. ORA-LEU-CAN study flow chart.**

Recurrence was defined as the reappearance of an OL at the site of surgery.

A clinical healthy mucosa had to be recorded with a clinical photograph between the time of surgery and recurrence (Fig 3).

All patients were given both written and verbal information about the study's objectives. Both a written and a verbal consent was obtained from the patients at the clinic before the inclusion. The written consent was signed by the patient and the clinician that included the patient in the study. The study was approved by the Regional Ethical Review Board in Gothenburg, Sweden (Dnr. 673–10) and was conducted in accordance with the Helsinki Declaration.

### Statistical analyses

Recurrence versus no recurrence of OL was the primary outcome. The follow-up time was defined as the time from the first excision surgery to the time of recurrence or to the last visit within the study protocol. In case of death before a recurrence, the death was censored (one patient).

The cumulative, disease-free survival rates were calculated using the Kaplan-Meier survival analysis, and the outcomes for patients in the different group were compared using a two-sided log-rank (Mantel-Cox) test. A recurrence was defined as an event in the Kaplan-Meier analysis and in the Cox regression analysis. Tests of the Cox proportional hazards assumption were conducted on the basis of Schoenfeld residuals after fitting a model.

**Table 1. Patients' characteristics.**

| | Patients N (%) |
|---|---|
| **Number of patients** | 180 (100.0) |
| **Gender** | |
| Male | 86 (47.8) |
| Female | 94 (52.2) |
| **Age (at inclusion) in years** | |
| Mean | 61 |
| Median | 62 |
| 20–29 | 3 (1.7) |
| 30–39 | 9 (5.0) |
| 40–49 | 11 (6.1) |
| 50–59 | 41 (22.8) |
| 60–69 | 69 (38.3) |
| 70–79 | 41 (22.8) |
| 80–89 | 5 (2.8) |
| 90–99 | 1 (0.5) |
| **Clinical diagnosis** | |
| Homogeneous | 109 (60.6) |
| Non-homogeneous | 71 (39.4) |
| **Histopathological diagnosis** | |
| Benign hyperkeratosis | 125 (69.5) |
| Lichenoid reaction | 20 (11.1) |
| Mild dysplasia | 15 (8.3) |
| Moderate dysplasia | 15 (8.3) |
| Severe dysplasia | 2 (1.1) |
| Verrucous hyperplasia | 3 (1.7) |
| **Site of lesion** | |
| Floor of the mouth | 10 (5.6) |
| Buccal mucosa | 23 (12.8) |
| Lateral tongue | 38 (21.1) |
| Ventral tongue | 13 (7.2) |
| Dorsum tongue | 3 (1.7) |
| Soft palate | 1 (0.5) |
| Hard palate | 14 (7.8) |
| Mandibular alveolar/gingival | 38 (21.1) |
| Maxillary alveolar/gingival | 32 (17.8) |
| Lip | 8 (4.4) |
| **Size of lesion** | |
| <200 mm$^2$ | 101 (56.1) |
| ≥200 mm$^2$ | 79 (43.9) |
| **Solitary or multiple OL** | |
| Solitary | 81 (45.0) |
| Multiple | 99 (55.0) |
| **Smoking** | |
| Yes | 109 (60.6) |
| No | 71 (39.4) |
| **Past history of smoking** | |

*(Continued)*

**Table 1.** (Continued)

|  | Patients N (%) |
|---|---|
| Yes | 70 (38.9) |
| No | 78 (43.3) |
| ND | 32 (17.8) |
| **Snuff** |  |
| Yes | 14 (7.8) |
| No | 166 (92.2) |
| **Past snuff use** |  |
| Yes | 25 (13.9) |
| No | 126 (70.0) |
| ND | 29 (16.1) |
| **Alcohol consumption** |  |
| Daily | 4 (2.2) |
| Several times per week | 20 (11.1) |
| Once a week | 60 (33.3) |
| Rarely/Never | 77 (42.8) |
| Never | 8 (4.5) |
| ND | 11 (6.1) |

*ND, No data available.*

To identify risk factors for predicting the recurrence of OL, a Cox regression analysis was utilised and the outcome was described as a Hazard Ratio (HR) with 95% confidence interval (CI). A P-value <0.05 was considered statistically significant. Statistical analyses were carried out using the SPSS Statistic for Macintosh ver. 25.0 software package (IBM Corp., Armonk, NY).

## Results

Overall, 103 (57%) of the total of 180 patients underwent surgical excision *in toto*, including 52 women and 51 men (mean age, 61 years; median age, 62 years; age range, 28–81 years) (Table 1). The median follow-up time to recurrence was 1.3 years (min–max: 0.2–5.7 years) and to the end of follow-up for those censored it was 4.0 years (min–max: 0.9–6.0 years).

Sixty-two patients (60%) were diagnosed with homogeneous OL, and 41 patients (40%) had non-homogeneous OL. Of the 103 patients with OLs that were excised, 43 lesions (42%) recurred. The cumulative incidence of recurrence after 4 years was 45% (95% CI, 35%–56%) and after 5 years was 49% (95% CI, 28%–60%). Twenty-four (56%) of the patients were men and 19 (44%) were women (Table 2).

Twenty (32%) recurrences were observed in the group with homogeneous OL (N = 62) and 23 (56%) in the group with non-homogeneous OL (N = 41). Non-homogeneous OL recurred more frequently than homogeneous OL (P = 0.021) (Table 2 and Fig 4A). A clinical diagnosis of non-homogeneous OL was associated with a 2-fold higher risk of developing a recurrence (95% CI,1.10–3.65; P = 0.024) (Table 3).

Seventeen patients (50%) in the recurring OL group had a lesion size >200 mm$^2$, as compared to 17 patients (50%) in the non-recurring group (Table 2). No significant difference was found between lesion size and recurrence (Fig 4B, P = 0.45).

Different anatomical sites in the oral cavity for the OL lesion showed different frequencies of recurrence. Recurrence was seen in 18 (50%) patients with OL on the tongue, 16 patients

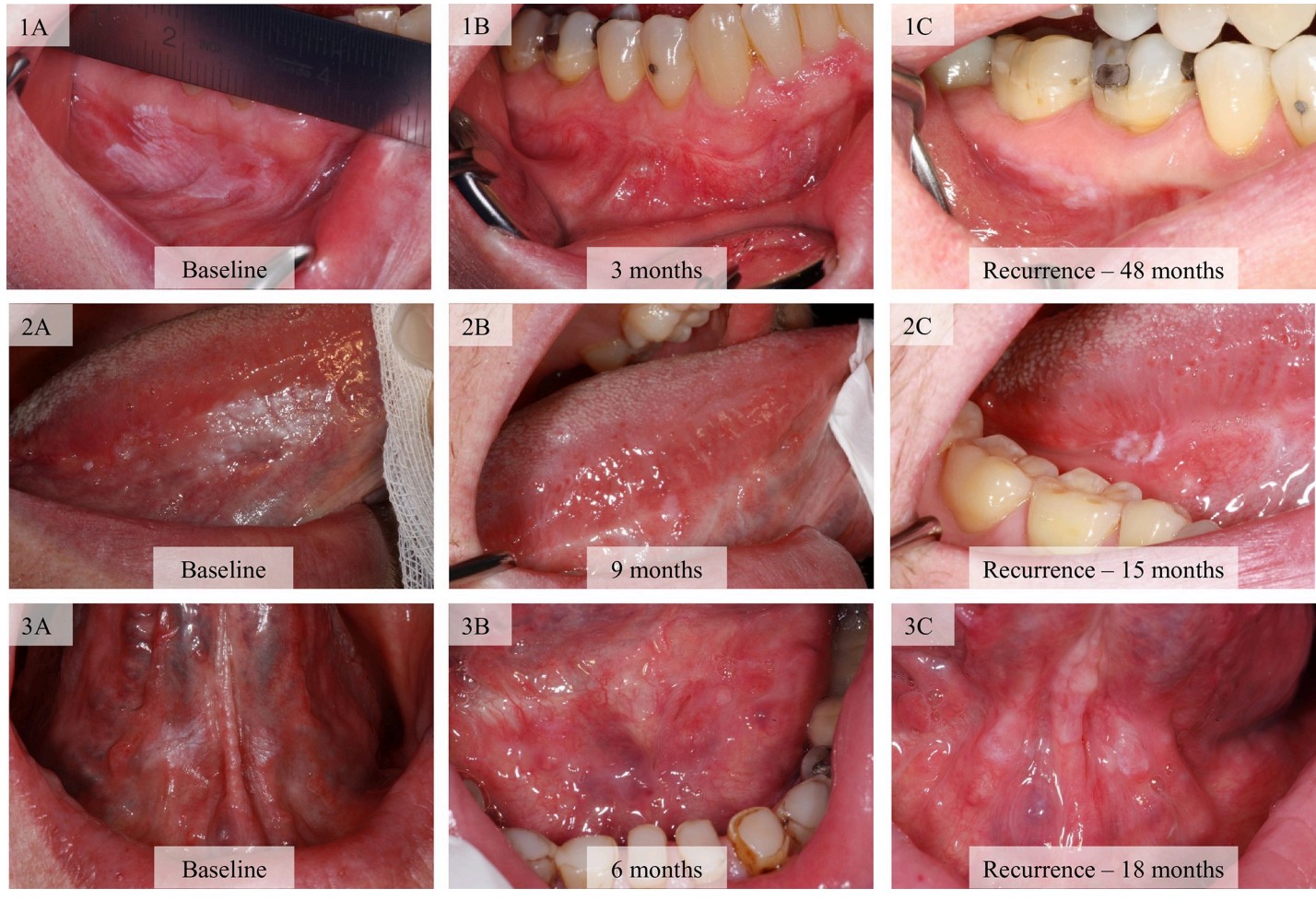

**Fig 3. Clinical appearances of three patients and the definition of a recurrence of leukoplakia.** Patient 1: A, B, C; 2: A, B, C; 3: A, B, C. A clinical healthy mucosa had to be recorded with a clinical photograph between the time of surgery and the time of recurrence.

(42%) with gingival OL, 3 (30%) patients with OL on the hard palate, 2 patients (50%) with OL in the floor of the mouth, 3 patients (27%) with OL in the buccal mucosa, and one patient (33%) with OL on the lip. We performed a site-recurrence analysis utilising a log-rank test and grouping the site of the lesion as: i) attached to the gingiva, including the hard palate: ii) the tongue; and iii) the buccal mucosa/floor of the mouth. No statistically significant correlation was found between the site of the lesion and recurrence (P = 0.34) (Table 2).

Twenty-four patients (47%) in the recurring group and 27 patients (53%) in the non-recurring group had multiple OLs. No significant difference was found between multiple lesion sites and the recurrence of OL (P = 0.59) (Table 2; Fig 4D).

In the cases of OL with dysplasia (N = 22), 12 (55%) recurred, while in the OL group without dysplasia (N = 81), 31 (38%) recurred (Table 2). No statistically significant difference in recurrence was detected between cases of OL with and without dysplasia (P = 0.17) (Table 2 and Fig 4C).

Among the 19 patients who smoked at inclusion, five (26%) had an OL that recurred (Table 2). No statistically significant difference in recurrence was detected between smokers and non-smokers (Fig 4E, P = 0.22).

**Table 2. Patients treated with surgical removal of leukoplakia.**

| | No recurrence | Recurrence | Total | P-value |
|---|---|---|---|---|
| | N (%) | N (%) | N | |
| **Patients** | 60 (58) | 43 (42) | 103 | |
| **Gender** | | | | NS |
| Male | 27 (54) | 24 (46) | 51 | |
| Female | 33 (63) | 19 (37) | 52 | |
| **Clinical diagnosis** | | | | 0.021 |
| Homogeneous | 42 (68) | 20 (32) | 62 | |
| Non-homogeneous | 18 (44) | 23 (56) | 41 | |
| **Size** | | | | NS |
| >200 mm$^2$ | 17 (50) | 17 (50) | 34 | |
| <200 mm$^2$ | 43 (62) | 26 (38) | 69 | |
| **Site of lesion** | | | | NS |
| Tongue | 18 (50) | 18 (50) | 36 | |
| Attached gingiva and hard palate | 29 (60) | 19 (40) | 48 | |
| Buccal mucosa and floor of the mouth | 13 (68) | 6 (32) | 19 | |
| **Number of lesions** | | | | NS |
| Multiple | 27 (53) | 24 (47) | 51 | |
| Single | 33 (63) | 19 (37) | 52 | |
| **Dysplasia** | | | | NS |
| Yes | 10 (45) | 12 (55) | 22 | |
| No | 50 (62) | 31 (38) | 81 | |
| **Smoker** | | | | NS |
| Yes | 14 (74) | 5 (26) | 19 | |
| No | 46 (55) | 38 (45) | 84 | |
| **Past smoker** | | | | NS |
| Yes | 18 (47) | 20 (53) | 38 | |
| No | 25 (60) | 17 (40) | 42 | |
| ND | 3 (75) | 1 (25) | 4 | |
| **Snuff use** | | | | 0.003 |
| Yes | 3 (27) | 8 (73) | 11 | |
| No | 57 (62) | 35 (38) | 92 | |
| **Past snuff use** | | | | NS |
| Yes | 11 (79) | 3 (21) | 14 | |
| No | 45 (59) | 31 (41) | 76 | |
| ND | 4 (31) | 9 (69) | 13 | |
| **Alcohol consumption** | | | | NS |
| Low to moderate | 51 (61) | 33 (39) | 84 | |
| Excessive use of alcohol | 4 (40) | 6 (60) | 10 | |
| ND | 5 (56) | 4 (44) | 9 | |

ND, No data available; NS, Not statistically different at P≥0.05

Among snuff-users, eight (73%) cases of OL recurred, as compared to three (27%) cases of non-recurring OL (Table 2). All patients used Swedish moist snuff (*snus*). OL diagnosed in snuff users were not located at the site of snuff application. The use of snuff was a significant pre-disposing factor for recurrence. The Cox regression analysis showed a significant difference in recurrence rate between patients who used snuff and patients who did not (P = 0.003)

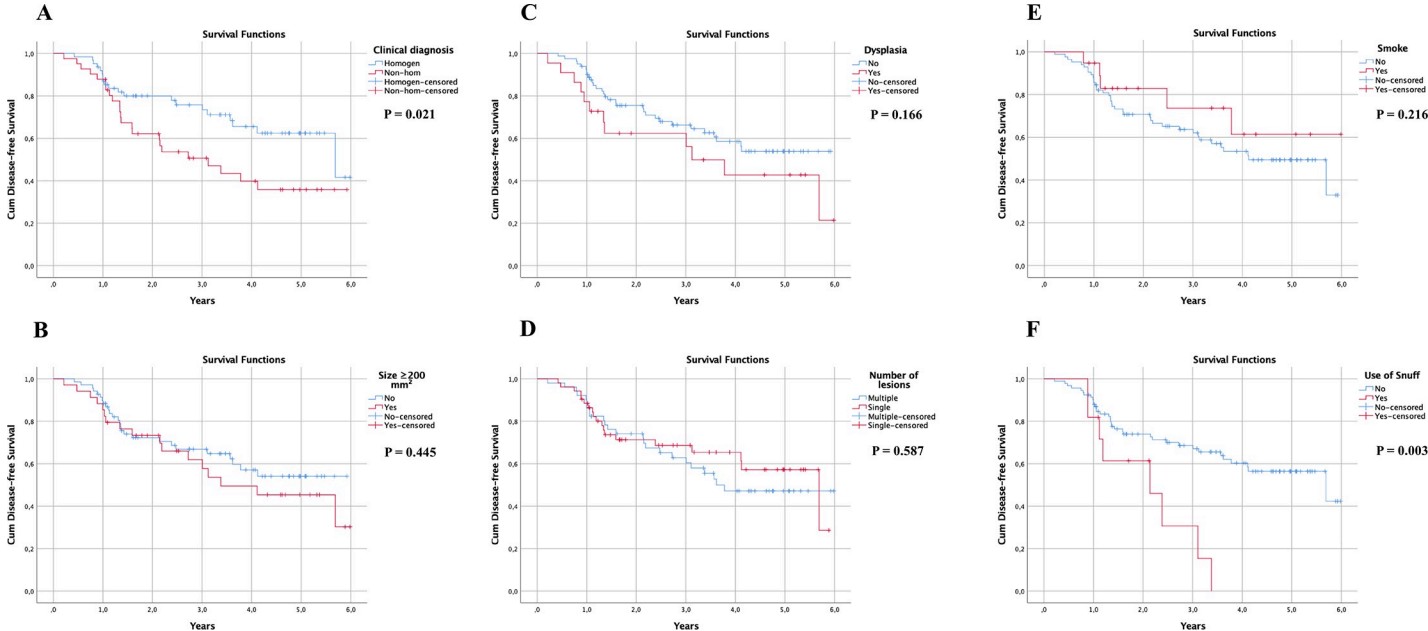

**Fig 4. Kaplan-Meier curves for disease-free survival of patients with OL.** Time shown is years from excision to recurrence. (A) Clinical diagnosis: P = 0.021; (B) Size: P = 0.445; (C) Dysplasia: P = 0.166; (D) Number of lesions: P = 0.587; (E) Smokers: P = 0.216; (F) Snuff users: P = 0.003.

(Fig 4F). The use of snuff was associated with a 3.11-fold increased risk of developing a recurrence (95% CI,1.41–6.86; P = 0.005) (Table 3).

A history of tobacco use, smoking or using snuff, did not show any significant correlation with the frequency of recurrence of OL (P = 0.22 and P = 0.13, respectively).

**Table 3. Cox regression analysis of risk factors for the recurrence of OL.**

|  | Uni-variable analysis | | Multi-variable analysis | |
|---|---|---|---|---|
|  | HR (95% CI) | P-value | HR (95% CI) | P-value |
| **Clinical diagnosis** |  |  |  |  |
| Homogeneous | 1.00 |  | 1.00 |  |
| Non-homogeneous | 2.00 (1.10–3.65) | 0.024 | 1.83 (1.00–3.37) | 0.052 |
| **Size** |  |  |  |  |
| <200 mm$^2$ | 1.00 |  |  |  |
| ≥200 mm$^2$ | 1.27 (0.69–2.35) | 0.45 |  |  |
| **Dysplasia** |  |  |  |  |
| Yes | 1.00 |  |  |  |
| No | 1.60 (0.82–3.13) | 0.17 |  |  |
| **Number of lesions** |  |  |  |  |
| Multiple | 1.00 |  |  |  |
| Single | 1.18 (0.65–2.16) | 0.59 |  |  |
| **Smoker** |  |  |  |  |
| Yes | 1.63 (0.64–4.15) | 0.30 |  |  |
| No | 1.00 |  |  |  |
| **Snuff use** |  |  |  |  |
| Yes | 3.11 (1.41–6.86) | 0.005 | 2.73 (1.22–6.08) | 0.014 |
| No | 1.00 |  | 1.00 |  |

The tobacco cessation counselling that all patients received according to the standard of care, resulted in 3/19 smokers stopping smoking, and none of these patients developed a recurrence. In contrast, 5 out of 16 patients who continued smoking developed a recurrence.

Among the 14 snuff-using patients, 4 quit using snuff and 3 of them developed recurrence despite tobacco cessation. Six out of the 8 patients who continued using snuff suffered a recurrence of their OL.

No significant correlation was found between the use of alcohol and recurrence of OL.

We used an alcohol-recurrence analysis that grouped the use of alcohol into the following groups: low/moderate use (including never used alcohol, rarely/never used, and intake of alcohol once a week); and excessive use of alcohol (including daily intake of alcohol and intake of alcohol several times per week). No significant difference in recurrence-free time was detected between the groups (P = 0.075) (Table 2).

Age and gender did not show any significant correlations with recurrence of OL (P = 0.62).

The results from the uni-variable and multi-variable Cox regression analyses revealed that the clinical diagnosis of non-homogeneous OL, showed a tendency to a correlation and the use of snuff remained as a significant risk for recurrence of OL (Table 3). The Cox proportional hazards assumption was fulfilled for all the analysed variables.

While no patient in the non-recurring OL group had a lesion that transformed into OSCC, four patients in the group with recurring OL developed OSCC (P<0.001).

## Discussion

The present study shows that a high proportion (42%) of OL cases recurred despite complete surgical removal. While this is in line with the results reported by Brouns et al. [11], we found higher recurrence rates than those reported by other groups [13, 19, 21]. However, in the latter studies, laser surgery was used instead of conventional scalpel surgery, which might have had an impact on the outcome. However, the present study shows a recurrence rate that exceeds that previously reported by Holmstrup et al. following scalpel surgery [15].

This study also shows that a non-homogeneous type of OL and the use of snuff predispose the patient to OL recurrence. In contrast, factors such as lesion size, dysplasia, site of the lesion, smoking, use of alcohol, the presence of multiple/single lesions, and gender do not differ significantly between cases of recurring and non-recurring OL. Our results showing that patients with non-homogeneous OL have a higher risk for recurrence after surgery than patients with homogeneous OL is in line with the findings reported in several studies [13, 21].

In the recurring OL group, 9% of the cases transformed into OSCC. This is in line with the 12% cancer transformation rate for OL reported by Holmstrup et al. in a retrospective study with a mean follow-up period of 7.5 years [15]. In the present study, patients with OL that transformed into OSCC experienced recurrence before OSCC transformation, leading to a significantly higher risk for malignant transformation for the patients with recurring OL than for those with non-recurring OL. This finding supports the results reported from earlier studies showing that despite surgical removal of the OL the risk for cancer transformation is not eliminated [11, 15]. Recurrence probably indicates the emergence of instability of the intracellular cell cycle-regulating mechanisms. Stringent regulation of cell cycle/cell division is a prerequisite for preventing cancer transformation [22].

The concept of 'field cancerization' is widely accepted as an explanation for the recurrence of OSCC after surgery [23, 24]. Field cancerization entails that there is a genomic instability throughout the epithelium that eventually can give rise to genomic aberrations in keratinocytes by random mutations at any site, leading to cancer. This concept could also be applied to OL, since genomic instability is present in the OL epithelium [8]. This makes a comparison

between patients with solitary OL and multiple OLs of interest. The latter may have more extensive genetic aberrations in the oral epithelium, thereby increasing their risk for OL. However, we observed no difference in recurrence rate between patients with solitary lesions and those with multiple lesions. Thus, in the present study, the presence of multiple OLs does not appear to be a useful marker of recurrence.

Kuribayashi et al have reported a significant correlation between surgical margins and recurrence of OL after surgery [19]. Biopsy specimens from patients in the present study were not subjected to serial sectioning. However, surgical margins were evaluated during the surgery by experienced specialists in maxillofacial surgery and in oral medicine. Clinically healthy mucosa was also verified at the follow-up visit, since the criterion for defining recurrence was that the clinical assessment also included a photograph of the lesion. While the absence of serial sections is a draw-back, even if serial sections are made this does not guarantee normal genomic regulatory mechanisms in the margins [24]. Determining the margins of OL during surgery remains a challenge, and optical adjuncts may be one way to address this. Tiwari et al. carried out a systemic review of the efficacy of direct optical fluorescence imaging as a supplement to complete oral examination in the clinical evaluation and surgical management of OL, and they concluded that the technique holds promise and may be used as a clinical adjunct in the treatment of OL [25]. This may answer the question as to where the margins of OL are located, and may provide a guideline for the surgical procedure.

In seven patients with OL the histopathological diagnosis was lichenoid reaction. In these patients, records and clinical photos were thoroughly re-reviewed. No clinical signs according to van der Meij et al [26], were found indicating oral lichen planus, oral lichenoid lesion or lichenoid contact reaction. A diagnosis of OL according to WHO [1] postulates exclusion of known diseases or disorders that carry no increased risk of cancers. In the present study we strictly adhered to this definition. In the WHO definition of OL, histopathological diagnosis is confined to presence of dysplasia or not [1]. Thus, excluding known diseases causing a white plaque in the oral mucosa, the histopathological diagnosis lichenoid reaction is compatible with an OL diagnosis.

Tobacco cessation counselling resulted in only 16% of the patients quitting smoking. Importantly, recurrence was not noted in any of patients who stopped smoking and in one-third of the patients who continued smoking. This emphasises the need for more-effective tobacco cessation programs [27, 28].

A limitation of the present study is that our cohort of patients is not a representative sample of the Swedish population. High-risk patients do not attend routine dental care to the same extent as the patients in this cohort, and this has to be considered when interpreting the results. Both OL and OSCC are over-represented in patients with excessive use of tobacco and alcohol [29]. Thus, the present study may well under-estimate the frequencies of both recurrence and OSCC transformation.

In conclusion, in this study, the cumulative incidence of recurrence of OL is found to be 45% after 4 years and 49% after 5 years. Parameters that predict the recurrence of OL are non-homogeneous clinical type and the use of snuff. There are no significant differences in recurrence between OL with or without dysplasia, lesion size, multiple OL vs. solitary OL, sites of the lesions, alcohol consumption or smoking. OL that recurs has a significantly higher risk of transforming into OSCC. Further studies are needed to identify biomarkers that can be used to predict the risk of recurrence after surgery, as well as the risk of cancer transformation.

## Acknowledgments

The authors also acknowledge the efforts of all our collaborators at the participating centres in recruiting patients to the ORA-LEU-CAN study. We also acknowledge Dr Vincent Collins for English proof reading of the manuscript.

## Author Contributions

**Conceptualization:** Jonas Sundberg, Mats Wallström, Jenny Öhman, Lars Sand, Jan-Michaél Hirsch, Göran Kjeller, Bengt Hasséus.

**Data curation:** Jonas Sundberg, Magdalena Korytowska.

**Formal analysis:** Jonas Sundberg, Erik Holmberg, Anikó Kovács.

**Funding acquisition:** Jonas Sundberg, Bengt Hasséus.

**Investigation:** Magdalena Korytowska, John Bratel, Mats Wallström, Ebba Kjellström, Johan Blomgren, Anikó Kovács, Jenny Öhman, Lars Sand, Jan-Michaél Hirsch, Daniel Giglio, Göran Kjeller, Bengt Hasséus.

**Methodology:** Mats Wallström, Lars Sand, Jan-Michaél Hirsch, Daniel Giglio, Göran Kjeller, Bengt Hasséus.

**Writing – original draft:** Jonas Sundberg, Jenny Öhman, Bengt Hasséus.

**Writing – review & editing:** Jonas Sundberg, Magdalena Korytowska, Erik Holmberg, John Bratel, Mats Wallström, Ebba Kjellström, Johan Blomgren, Anikó Kovács, Jenny Öhman, Lars Sand, Jan-Michaél Hirsch, Daniel Giglio, Göran Kjeller, Bengt Hasséus.

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
