## [Decision Letter · Decision Letter 0]

24 Oct 2019

PONE-D-19-26374

Recurrence rates after surgical removal of oral leukoplakia - a prospective longitudinal multi-centre study

PLOS ONE

Dear Dr. Hasséus,

Thank you for submitting your manuscript to PLOS ONE. After careful consideration, we feel that it has merit but does not fully meet PLOS ONE’s publication criteria as it currently stands. Therefore, we invite you to submit a revised version of the manuscript that addresses the points raised during the review process.

We would appreciate receiving your revised manuscript by Dec 08 2019 11:59PM. To enhance the reproducibility of your results, we recommend that if applicable you deposit your laboratory protocols in protocols.io, where a protocol can be assigned its own identifier (DOI) such that it can be cited independently in the future. For instructions see: http://journals.plos.org/plosone/s/submission-guidelines#loc-laboratory-protocols

We look forward to receiving your revised manuscript.

Kind regards,

Muy-Teck Teh, Ph.D.

Academic Editor

PLOS ONE

Journal Requirements:

2.  Please include the full name of the IRB/ethics committee that reviewed and approved this study, including the name of the affiliated institution if applicable. We additionally ask that you include your IRB/ethics committee approval number in your ethics statement.

For additional information about PLOS ONE ethical requirements for human subjects research, please refer to " ext-link-type="uri" xlink:type="simple">http://journals.plos.org/plosone/s/submission-guidelines#loc-human-subjects-research."

3.  We suggest you thoroughly copyedit your Abstract for language usage, spelling, and grammar. If you do not know anyone who can help you do this, you may wish to consider employing a professional scientific editing service.  

4. Please provide additional details regarding participant consent in the Methods section of your manuscript. Please ensure that you have specified (1) whether consent was informed and (2) what type you obtained (for instance, written or verbal, and if verbal, how it was documented and witnessed). If your study included minors, state whether you obtained consent from parents or guardians.

5. In your Methods section, please provide additional information about the participant recruitment method and the demographic details of your participants. Please ensure you have provided sufficient details to replicate the analyses such as: a) the recruitment date range (month and year), e) a description of how participants were recruited, and f) descriptions of where participants were recruited  and where the research took place (institution names of the participating centres).

6. 

We note that you have indicated that data from this study are available upon request. PLOS only allows data to be available upon request if there are legal or ethical restrictions on sharing data publicly. For information on unacceptable data access restrictions, please see http://journals.plos.org/plosone/s/data-availability#loc-unacceptable-data-access-restrictions.

Reviewers' comments:

Reviewer's Responses to Questions

**Comments to the Author**

1. Is the manuscript technically sound, and do the data support the conclusions?

Reviewer #1: Yes

Reviewer #2: Yes

2. Has the statistical analysis been performed appropriately and rigorously? 

Reviewer #1: Yes

Reviewer #2: I Don't Know

3. Have the authors made all data underlying the findings in their manuscript fully available?

Reviewer #1: Yes

Reviewer #2: Yes

4. Is the manuscript presented in an intelligible fashion and written in standard English?

Reviewer #1: Yes

Reviewer #2: Yes

5. Review Comments to the Author

Reviewer #1: The authors have carried out a very interesting study to determine the causes for recurrence of oral Leukoplakias

The study has been carried out in a very systematic manner,

Oral leukoplakia with dysplasia are potentially premalignant lesions. In the current study the number of cases with dysplasias that is selected are only 22. More number of cases with dysplasias can be evaluated and the recurrence rate and causes for recurrences for these cases should be determined. This will help in identifying high-risk lesions and they can be treated appropriately.

The Kaplan - Meier survival curves are also very interesting. It is interesting to note that the habit of snuff is a causative factor for recurrence.

Reviewer #2: This is an interesting and well performed study on recurrence of oral leukoplakias (OL) following surgical excision. I have only three comments:

1. A definition of oral leukoplakia is given in the introduction, however, I think the definition used by the participating centers should be given also in the patients and methods section.

2. On page 4/5 it is stated that “the inclusion criterion was a clinically and histopathologically verified diagnosis of OL”. Traditionally, histopathology cannot verify an OL, however, the biopsy can rule out other diagnoses and give information on possible epithelial dysplasia/carcinoma. I am sure the authors are aware of this, however, in order not to mislead others, I think the sentence should be changed.

3. In Table 1 lichenoid reaction is given as the histopathological diagnosis in 20 cases. Three questions emerge from this:

a. is this compatible with a final diagnosis of leukoplakia? A short discussion could be included.

b. lichenoid reactions are sometimes seen in relation to contact with dental restorations or other causes. Were there any clinical indications that the cases with histopathologically diagnosed lichenoid reactions had an obvious (or possible) cause such as contact with dental restorations? I think a short mentioning of this should appear in the patients and methods section.

c. how many recurrences were seen in these cases?

6. PLOS authors have the option to publish the peer review history of their article (what does this mean?). If published, this will include your full peer review and any attached files.

Reviewer #1: Yes: Dr Monica Charlotte Solomon

Reviewer #2: No

---

## [Author Response · Author response to Decision Letter 0]

4 Nov 2019

Response to the reviewers´ comments, 

Thank you very much for valuable comments. We are much obliged for your thorough revision. Please find our response in italics below. 

Please find a point-by-point response to the Reviewers comments below. 

Reviewer # 1. 

“Oral leukoplakia with dysplasia are potentially premalignant lesions. In the current study the number of cases with dysplasias that is selected are only 22. More number of cases with dysplasias can be evaluated and the recurrence rate and causes for recurrences for these cases should be determined. This will help in identifying high-risk lesions and they can be treated appropriately.”

We agree. But the 22 cases with dysplasia are the patients to date available in this prospective study. However, the ORA-LEU-CAN study is ongoing and we hope in the future to be able to answer the issues raised by the referee.

 Reviewer #2. 

1) “A definition of oral leukoplakia is given in the introduction, however, I think the definition used by the participating centers should be given also in the patients and methods section#. 

The definition of OL has been included in the Patients and Methods section (P.5 para.1). 

2) “…the inclusion criterion was a clinically and histopathologically verified diagnosis of OL” is correct. 

We agree that OL is a solely clinical diagnosis. The sentence has been rephrased:

“the inclusion criterion was a clinically verified diagnosis of OL” (p.5, para. 1). 

3) In Table 1 lichenoid reaction is given as the histopathological diagnosis in 20 cases.

Table 1 illustrates the patients’ characteristics in the entire patient group (N = 180) but we analysed only the 103 patients who had an OL that we were able to be completely remove. Of these seven patients four recurred.

Three questions emerge from this:

a. Is this compatible with a final diagnosis of leukoplakia? A short discussion could be included.

We have elucidated this matter in the Discussion (p.12, para.2):

“In seven patients with OL the histopathological diagnosis was lichenoid reaction. In these patients, records and clinical photos were thoroughly re-reviewed. No clinical signs according to van der Meij et al (26), were found indicating oral lichen planus, oral lichenoid lesion or lichenoid contact reaction. A diagnosis of OL according to WHO (1) postulates exclusion of known diseases or disorders that carry no increased risk of cancer. In the present study we as strictly adhered to this definition. In the WHO definition of OL, histopathological diagnosis is confined to presence of dysplasia or not (1). Thus, excluding known diseases causing a white plaque in the oral mucosa, the histopathological diagnosis lichenoid reaction is compatible with an OL diagnosis.”

b. Lichenoid reactions are sometimes seen in relation to contact with dental restorations or other causes. Were there any clinical indications that the cases with histopathologically diagnosed lichenoid reactions had an obvious (or possible) cause such as contact with dental restorations? I think a short mentioning of this should appear in the patients and methods section.

None of the seven patients had dental fillings in contact with the lesions. A sentence describing this has been added in the manuscript (P.x, para y) In that group seven patients had the histopathological diagnosis lichenoid reaction.

c. How many recurrences were seen in these cases

4 OL out of 7 with the histopathological diagnosis lichenoid reaction recurred.

Yours sincerely,

Bengt Hasséus, LDS, PhD, Associate Professor

Corresponding author

Dept. of Oral Medicine and Pathology,

Institute of Odontology, Sahlgrenska Academy,

University of Gothenburg,

PO Box 450, 

SE 405 30 Gothenburg, Sweden

Email: bengt.hasseus@gu.se

---

## [Editor Report · Decision Letter 1]

12 Nov 2019

Recurrence rates after surgical removal of oral leukoplakia - a prospective longitudinal multi-centre study

PONE-D-19-26374R1

Dear Dr. Hasséus,

We are pleased to inform you that your manuscript has been judged scientifically suitable for publication and will be formally accepted for publication once it complies with all outstanding technical requirements.

With kind regards,

Muy-Teck Teh, Ph.D.

Academic Editor

PLOS ONE
---

## [Editor Report · Acceptance letter]

22 Nov 2019

PONE-D-19-26374R1 

Recurrence rates after surgical removal of oral leukoplakia - a prospective longitudinal multi-centre study 

Dear Dr. Hasséus:

I am pleased to inform you that your manuscript has been deemed suitable for publication in PLOS ONE. Congratulations! Your manuscript is now with our production department. 

With kind regards,

on behalf of

Dr. Muy-Teck Teh 

Academic Editor

PLOS ONE